# Microstructure and Properties of CoCrNi/Nano-TiC/Micro-TiB2 Composite Coatings Prepared via Laser Cladding

**DOI:** 10.3390/ma16217016

**Published:** 2023-11-02

**Authors:** He Liu, Yuzhen Yu, Xi Wang, Hanpeng Gao, Jinku Zhao, Hao Wang

**Affiliations:** 1School of Mechanical Engineering, Yancheng Institute of Technology, Yancheng 224051, China; icecream20231031@163.com (H.L.); zhaojinku19980121@163.com (J.Z.); rwang2814@gmail.com (H.W.); 2School of Electrical Engineering, Yanshan University, Qinhuangdao 066004, China; gaohanpeng@ysu.edu.cn

**Keywords:** laser cladding, grain refinement, diffusion reinforcement, wear mechanisms

## Abstract

Laser cladding was used to prepare CoCrNi-xTiC-xTiB2 (x = 0, 5, 15 wt.%) composite coatings on 316L stainless steel. Then, ceramic mass fraction effects on the microstructure and properties were investigated. Results show viable metallurgical bonding between the coating and the substrate, with no apparent pores or cracks. The addition of ceramics transformed the coating phase from a single-phase face-centered cubic (FCC) to a multi-phase FCC+TiC+TiB2. TiC and TiB2 increased the hardness of the CoCrNi-xTiC-xTiB2 coating from 209.71 HV to 494.77 HV by grain refinement and diffusion strengthening. The substrate wear loss was 0.0088 g, whereas the CoCrNi-xTiC-xTiB2 (x = 15%) coating wear loss was only 0.0012 g. Moreover, the overall wear mechanism of the coating was changed: the substrate wear mechanism was used for abrasive wear, adhesive wear and fatigue wear, and the coating with the addition of 15 wt.% nano-TiC and 15 wt.% micro-TiB2 was the wear mechanism for pitting fatigue wear.

## 1. Introduction

The type of stainless steel that is widely used in aerospace engineering, biomedical systems, smart manufacturing and transportation is 316L stainless steel due to its excellent mechanical properties, low coefficient of thermal expansion, and good corrosion resistance [1,2,3,4]. However, it possesses low surface hardness and poor wear resistance [5,6]. In order to broaden the range of applications and extend the service life of 316L steel, a reinforced layer is produced onto the relevant metal surface. Laser cladding using a high-energy laser beam can quickly fuse the powder and the substrate through the formation of a cladding layer on the substrate surface. The method has various advantages, among which are low dilution rate, small heat-affected zone, fast solidification speed, large energy density, and good metallurgical bonding with the substrate [7,8,9,10].

The materials currently applied in laser cladding are nickel-based, cobalt-based, and iron-based matrix composites [11,12,13,14]. In recent years, medium-entropy alloys—a type of alloy composed of three to four elements with similar molar ratios—have attracted much attention from researchers due to their excellent properties [15]. The structure of medium-entropy alloys is usually a relatively simple solid solution with a body-centered cubic, face-centered cubic or densely packed hexagonal lattice. Compared with high-entropy alloys, medium-entropy ones have comparable performance and lower cost, which is of great application significance. One typical medium-entropy alloy is CoCrNi which possesses good ductility, high yield strength, outstanding wear resistance, and excellent fracture toughness [16,17,18]. For instance, Liu et al. [19] prepared CoCrNi coatings in rain gauge shells via laser cladding technology to study the coating structure and properties. The results showed that the CoCrNi coating exhibited a smaller corrosion current density than 304 stainless steel in solutions with different PH values. The corrosion resistance, hardness and wear resistance of this coating were also greatly improved. Ma et al. [20] produced metal-ceramic composite coatings by adding 5 wt.%, 15 wt.% and 25 wt.% WC to CoCrNi powder so as to monitor the evolution of carbides and the effect of synergism on the coatings. It was found that the coating dendrites were continuously refined with the increase in WC content. Moreover, the microhardness of the 25 wt.% WC coating increased by 162.46% relative to that of the pure CoCrNi coating due to strengthening in both solid solution and second phase structures. In addition, the increase in the hardness of the coating reduced the influence of the wear friction so that the coefficient of friction of the coating fell from 0.70 to 0.52. Chen et al. [21] used laser cladding to prepare TiC and TiB2 in situ and investigated their microstructure and properties in iron-based coatings. Results suggest that the Ti element can improve the surface flatness of the coating, whilst the formed TiC and TiB2 refined the grains, thereby increasing their microhardness. Furthermore, adding Ti and B4 C changed the wear mechanism from adhesive to abrasive wear. Wang et al. [22] performed the ultrasonic shock treatment using different shock amplitudes to improve the coating properties during the laser cladding process. It was found that the grain size before the ultrasonic treatment was about 85.847 μm, reduced to about 64.067 μm (by 34%) after the exposure to a 20 μm impact amplitude. The microhardness of the treated coating was also significantly improved (by about 2.22 times) relative to that of the initial coating, whereas the wear weight was only 14.3% of that of the untreated coating. Zhuang et al. [23] used laser cladding to prepare TiC-reinforced CrMnFeCoNi coatings and investigate wear properties at room temperature and 873 K. Results indicate that TiC could significantly increase the microhardness of the coatings and thus improve wear resistance. Increasing TiC content at room temperature reduces the fluctuation of the coating friction coefficient. In addition, higher TiC content resulted in wear-prone areas due to uneven TiC distribution; at a 873 K temperature, the actual wear area temperature was lower than the phase-stable TiC temperature, which improved wear resistance. In addition, plasma sintering [24], vacuum arc melting [25] and stir friction welding [26] have been used to manufacture high-performance CoCrNi alloys. Meanwhile, most works have focused so far on the preparation of single CoCrNi coatings or CoCrNi alloys under varying process parameters, while there are relatively few studies on introducing the second phase into CoCrNi to prepare metal-ceramic composite coatings and improve their properties.

Metal-based ceramic materials combine the advantages of both types of constituents, such as good wettability of metals, high hardness of ceramics, better wear resistance and other properties. TiC and TiB2 are ideal reinforcing phases due to their high hardness, low coefficient of friction and excellent chemical stability. It is worth noting that the size of ceramic particles also affects the performance of the whole coating. In particular, the refinement effect of nanoparticles imposed on grains is better than that of microparticles. In turn, the wear rate of the microparticle-composed coating is lower than that of the coating based on nanoparticles, and the relevant wear resistance is also better. Therefore, two different types of ceramics with varying particle size were used in this work to prepare metal–ceramic composite coatings on 316L stainless steel substrates by adding appropriate ratios of nano-TiC, micro-TiB2 and CoCrNi powders under continuous wave laser exposure. Particular attention was paid to the influence of additives on the microstructure, hardness and wear resistance of the coatings.

## 2. Experimental

### 2.1. Coating Preparation

The substrates used in this study were made of commercially available 316L stainless steel (Taizhou Qiuxin Metal Products Co., Ltd., Taizhou, China) and had the dimensions of 50 mm × 50 mm × 10 mm. Substrates were polished with sandpaper to remove surface oil, ultrasonically cleaned with alcohol, and dried prior to the experiment. The cladding materials were CoCrNi (Xindun Metal Materials Co., Ltd., Xindun, China), nano-TiC (99.99%, 40 nm, Xindun Metal Materials Co., Ltd., Xindun, China) and micro-TiB2 (99.9%, 1 μm, Xindun Metal Materials Co., Ltd., Xindun, China). The powders were weighed according to the required proportions and exposed to ball milling for 3 h. The SEM images of the powders are shown in Figure 1a–c. The powders were spread on the substrate to a thickness of 1 mm and then exposed to the laser fusion by means of a Raycus RFL-A2000 D fiber laser (Model-A2000D fiber laser; Make-Raycus RFL, Wuhan, China) conforming to the parameters listed in Table 1. The cladding process was controlled by the ROKAE XB16 six-axis robot (Model-XB16 six-axis robot; Make-ROKAE, Beijing, China) using argon as the shielding gas. The schematic diagram of the experimental equipment is illustrated in Figure 1d. The corresponding numbers of coatings and their heights are given in Table 2.

### 2.2. Testing Method

Samples with dimensions of 10 mm × 10 mm × 10 mm were cut using a wire cutting machine and then set with a metallographic setting machine. After that, specimens were exposed to sanding and polishing with 360#, 600#, 1000#, 1500#, 2000# and 5000# sandpapers, respectively. At last, the mechanical polishing with W0.5 diamond paste was performed until the sample surface was free of scratches, and the obtained coating morphology was observed in the optical microscope (Model-M3LY630T; Make-Shunyu, Holland). Each coating was immersed in a metallographic etching solution for 6 min to initiate corrosion and then rinsed with alcohol. A scanning electron microscope (Model-Nova Nano SEM450; Make-FEI, Hillsboro, OR, USA) was used to examine the coating structure, and an energy dispersive spectrometer (Model-VANTAGE-DS1; Make-NORAN, Bloomington, MN, USA) was employed to measure the elemental distribution and content of the coating in the specified area. An X-ray diffractometer (Model-X PERT3 POWDER; Make-PANalytical B.V., Tokyo, Japan) was used to analyze the phase composition of the coatings over the 2θ range of 30° to 90°, and phase matching was carried out using MDI Jade 6 software. An HVS-1000 microhardness tester (Model-HVS-1000; Make-Yizong, Shanghai, China) was used to evaluate the microhardness of the coatings at 100 μm intervals along the vertical direction of the cross-section under a load of 100 g and a dwell time of 15 s. The wear performance of the coatings against the substrate was tested conforming to the linear reciprocating friction method using a CETR-UMT-3 MO Friction and Wear Tester(Model-UMT-3; Make-CERT, USA). The wear load was 50 N, the frequency was 2 Hz, the single stroke was 5 mm and the wear time was 30 min. The wear parts were 4 mm diameter GCr15 steel balls. The change in coefficient of friction was automatically recorded by a wear tester sensor system. Before the experiment, the coatings were ground to bring their thickness to half of the initial value and then polished to avoid the impact of surface roughness on the friction coefficient. The coating wear traces were observed in a scanning electron microscope. A VK-X110 shape measuring laser microscope (Model-VK-X110; Make-KEYENCE, Shanghai, China)was applied to characterize the three-dimensional wear morphology of the coatings; the abraded samples were ultrasonically cleaned with alcohol and then dried, and the weight of the samples was measured with an electronic balance accurate to 0.1 mg, and the weight of the abraded samples was calculated.

## 3. Results and Discussion

### 3.1. Macroscopic Morphology and Dilution Rate

Figure 2 shows the surface and cross-sectional morphology of the three groups of coatings. As can be seen, the coatings possessed the continuous and smooth surface without cracks, pits or any other defects. Moreover, there were no obvious pores and cracks in the cross-section, and the well-defined demarcation line between the fusion-coated area and the substrate highlighted the fusion-coated layer of good quality. With the addition of both TiC and TiB2, the cross-sectional parameters of the coating underwent the pronounced changes. In particular, the height of the coating without ceramic addition was about 730 μm and the depth was about 250 μm. Once the ceramic phase in a content of 30 wt.% was introduced, the height fell to about 345 μm and the depth increased to approximately 500 μm. It is noteworthy that the alterations in cross-sectional parameters exert a noticeable effect on the dilution rate. If the dilution rate is too high, the substrate melts more rapidly and the substrate elements penetrate into the fused cladding layer, which may reduce the performance of the latter. In turn, a very low dilution rate may result in poorer bonding between the substrate and the fused cladding layer or even fused cladding layer spalling [27]. To estimate the possible risks associated with coating quality, the dilution rates of the coatings were further calculated using the formula η = h/(h + H) [28], where h is the depth and H is the height of the coating. The corresponding values are shown in Table 3.

As can be seen from Table 3, the dilution rate of the 30 wt.% ceramic-coated specimen (59.16%) was more than twice that of the uncoated one (25.57%). This is mainly due to the fact that during laser cladding, the cladding material absorbs energy and rapidly forms a molten pool and metal vapor. The metal vapor creates a back pressure that causes the unmolten powder particles to spray, resulting in poor powder utilization and a large reduction in coating height. In addition, the Marangoni effect is also an important factor in powder splashing [29,30]. Furthermore, under the same process parameters, the reduction in pre-positioned powder in the laser cladding increases the energy adsorption by the substrate and the temperature of the melt pool, which deepens the melt pool and has a greater impact on dilution rate.

### 3.2. Phase Composition

Figure 3 shows the XRD patterns of laser cladding of CoCrNi-xTiC-xTiB2 composite coatings with different ceramic contents. Co, Cr, and Ni are all present as single crystal structures in the hybrid powders. And as can be seen from Figure 3, the laser cladding caused the CoCrNi ternary alloy to form a single-phase face-centered cubic solid solution, and the phase of the TCB0 coating consists of a single FCC phase, which agrees with the previous reports [31,32], indicating that the high entropy effect of the multi-major elemental CoCrNi coatings inhibited the formation of intermetallic compounds. When TiC and TiB2 were added, the coating phase changed but retained the single-phase FCC structure of CoCrNi, which transformed from a single FCC phase to a multi-phase of FCC + TiC + TiB2 without forming other phases. This transformation suggests that, during the laser cladding process, the added TiC and TiB2 do not react with other elements due to excellent chemical stability, whilst the TiC and TiB2 phases are preserved in the coating, contributing to second-phase reinforcement. The TiC and TiB2 diffraction peaks were relatively weak when 10 wt. % of ceramics were added. The increase in the ceramic content to 30% strengthened the diffraction peaks, indicating that moderate addition of TiC and TiB2 in the pre-preg powder can create higher ceramic content in the laser cladding coatings, thus improving the microhardness and wear resistance of the coatings.

### 3.3. Microstructure

Figure 4 depicts the microstructure of the three sets of coatings. Figure 4(a1)–(c1) show the histological morphology of the bottom, middle and top of the TCB0 coatings. As seen from the figure, the TCB0 coating consists mainly of dendrites and equiaxed crystals. According to solidification theory, grain formation is mainly influenced by the temperature gradient (G) and the solidification rate (R) [33,34]. During the coating process, the temperature gradient at the substrate increases, whereas the solidification rate becomes smaller, promoting the increase in the G/R ratio and the formation of planar crystals on the substrate. Along the vertical direction of the coating cross-section, the temperature gradient gradually decreases while approaching the surface of the cladding layer, and the structure solidification is accelerated, thereby decreasing the G/R ratio. As a result, dendritic and equiaxed grains are precipitated separately. At the top of the cladding layer, the G/R ratio is the lowest, so the surface of the cladding layer exhibits equiaxed grains. In addition, the center of the light source in the laser cladding possesses the highest energy and heat, and there is the heat transfer from the center to the surroundings. Thus, the growth direction of the dendrites has a certain angle with the vertical direction, and the closer it is to the surface of the cladding layer, the lower the energy and the smaller the angle. Figure 4(a2)–(c3) display the microstructures of TCB10 and TCB30 coatings. As seen from the figure, the introduction of ceramic particles causes the refinement of the coating grains and the reduction in the dendritic features, so the coating grains become predominately equiaxed. Moreover, the higher the amount of ceramic particles, the smaller the grains. This is because the added ceramic particles can effectively inhibit grain growth. Figure 5 and Table 4 display the EDS surface scanning data on the three groups of coatings.

As shown in Figure 5a, the elements are uniformly distributed within the TCB0 coating with no apparent segregation. This is mainly due to the fact that laser melting ensures fast solidification, which reduces the element flow time. For the TCB10 and TCB30 coatings, the ceramic particles are mainly distributed at the grain boundaries, hindering grain growth, and the presence of Fe is detected in all three groups of coatings. The elemental exchange between the matrix elements and the cladding material also indicates that the cladding layer has a good metallurgical bonding with the substrate.

### 3.4. Microhardness

Figure 6 displays the microhardness change along the vertical direction of the cross-section of the coatings and the average hardness of the fusion coated zone. As seen from Figure 6a, the microhardness values can be divided into three regions, corresponding to the matrix zone, the heat-affected zone and the fusion cladding zone. The matrix has the lowest hardness, while the heat-affected zone has the second highest hardness value owing to the mutual diffusion of elements, and the cladding zone exhibits the highest hardness. The analysis of Figure 6a,b reveals that the hardness of the pure CoCrNi coating is relatively uniform, with an average value of about 209.71 HV, and the average hardness of the substrate is about 180.21 HV, which is a relative increase of about 16.37%. This is mainly because the matrix microstructure of 316L stainless steel is relatively coarse, while the CoCrNi coating produced by laser melting cladding has a uniform microstructure and small grains which are beneficial to the increase in hardness. Once ceramics are added, the microhardness undergoes an obvious increase. At the 10 wt.% ceramic content, the average microhardness is about 328.02 HV, exceeding that of the substrate by 82.02%. A further increase in the ceramic content to 30 wt.% results in the average microhardness of 494.77 HV, which is 174.55% higher than that of the substrate. The improvement of microhardness is due to the fine crystal strengthening, but also the diffusion strengthening. The melting point of TiC is 3150 °C [35], and the particle size is about 40 nm; therefore, the TiC particles are only partly decomposed upon laser melting and cladding, and their distribution in the coating is relatively uniform, hindering the movement of dislocations and thus increasing the hardness of the coating. As shown in Figure 6a, the higher the ceramic content, the more pronounced the diffusion strengthening effect. Meanwhile, TiB2 also experiences a noticeable grain strengthening [36]. In addition, the hardness of the coating increases due to the incorporation of hard phases. Since the dispersion of the hard phase in the coating is not completely uniform, it causes the fluctuation of the hardness within the coating.

### 3.5. Wear Resistance

Figure 7 shows the evolution in the coefficient of friction with wear time and the average coefficient of friction of the matrix and composite coatings with 0, 10 and 30 wt.% ceramic contents. The average friction coefficients of the matrix and the mentioned coatings were 0.5369, 0.6491, 0.6172 and 0.5950, respectively. It is noteworthy that even though the friction coefficients do not directly prove the wear resistance of the materials, they can reflect the change in wear of the materials at different moments. According to Figure 7a, the wear process required two stages whereby the friction coefficient first increased drastically and then gradually became smooth. The substrate and TCB0 coating entered the stable wear stage within 60 s, while the TCB10 and TCB30 coatings achieved the smooth stage after about 500 s, which was mainly determined by the nature of the material itself. In the stable wear stage, the fluctuation of the coefficient of friction was mainly due to the flaking accumulation of abrasive chips.

Figure 8 depicts the weight loss changes in the substrate and coatings during wear. As seen from the plot, the substrate lost the most weight (approximately 0.0088 g) among the specimens under consideration. In contrast, the coatings exhibited the lower weight loss. Among them, the more intense weight loss was recorded in the TCB0 coating (about 0.0082 g), while the coatings with added ceramic particles had the much slower reduction in weight (0.0038 g and 0.0012 g for the TCB10 and TCB30 coatings, respectively). Therefore, compared to the weight loss of the substrate, those coatings were reduced by 56.82% and 86.36%, respectively. Thus, the coatings with ceramic particles possess the better wear resistance than undoped ones.

Figure 9 shows the cross-sectional contours of the substrate and coating wear marks. As can be seen from the figure, the wear marks are related to the hardness of the material. In particular, the difference between the hardness values of the substrate and the TCB0 coating was small, and the width and depth of their cross-sectional profiles remained basically the same, with only a small variation in weight loss during wear. The addition of ceramics, on the other hand, resulted in a significant reduction in the width and depth of wear, especially when the ceramic content reached 30 wt.% (by about 25% of that of the substrate). In addition, the profile of TCB30 exhibited fewer bumps and pits and the relatively flat surface.

Figure 10 depicts the 3D wear morphology graphs of the substrate and coatings. Table 4 contains the parameters associated with the wear surface roughness. The wear degree of the substrate and coatings can also be seen in the figure. Under the same wear conditions, the wear of the substrate with TCB0 and TCB10 coatings was deeper compared to that of the TCB30 coating. In particular, the substrate with TCB0 coating exhibited severe plastic deformation and more intense wear. Moreover, according to the roughness parameters in Table 5, the average roughness values of the substrate and coatings were 31.009 μm, 29.878 μm, 17.700 μm, 3.244 μm, respectively. This indicates that increasing the hardness can effectively prevent plastic deformation of the coating during the wear process, which reduces the degree of abrasion of the material and prevents the flaking of the large-sized debris, thereby decreasing the coefficient of friction of the surface and improving the wear resistance.

The wear surface morphologies of the substrate and coatings are shown in Figure 11. As seen from Figure 11(a1,a2), there were plenty of uneven furrow-like features on the surface of the substrate, accompanied by many materials bonded together, indicating that the substrate was undergoing the abrasive and adhesive wear mechanisms. We recall that the hardness value of the substrate was low comparative to that of the counter-abrasive vice GCr15. During the wear process, the substrate experienced severe plastic deformation along with the adhesion of spalled debris to its surface, which resulted in the pile-up phenomenon and the formation of an adhesive layer, increasing the roughness of the surface. It can be seen in Figure 11(a2) that the wear marks had a few cracks, indicating that the substrate also suffered from fatigue wear. Figure 11(b1,b2) display the wear surface of the TCB0 coating. Compared to the substrate, the wear marks on the TCB0 coating were less pronounced, the depth of the groove was significantly reduced and lots of debris could be observed in the vicinity of the groove without any cracking. Abrasive and adhesive wear are still the main wear mechanisms. The addition of ceramic particles enables to noticeably improve the hardness of the coating, which plays a positive role in wear resistance. According to Figure 11(c1,c2), the wear marks on the coating were shallow, the plough-like feature almost disappeared, and a small amount of abrasive debris was present on the coating surface, meaning that adhesive wear was the predominant wear mechanism. In the case of the TCB10 coating with a 10 wt.% ceramic content, the hard phase particles effectively resisted the plastic deformation during the abrasion process, which improved the abrasion resistance of the entire material. Besides that, the debris and material bonded to the coating surface were mainly attributed to the CoCrNi powder with the low hardness, which would have cut off first upon the whole abrasion process. Once the ceramic content reached 30%, the coating resisted GCr15 cutting ability more. Meanwhile, the wear degree was shallowest and the wear surface revealed neither the furrow-like features nor the material bonding. Meanwhile, the increase in hard phase content promoted the emergence of pitting fatigue wear with the smaller local areas of material loss on the coating surface under contact pressure [33]. As shown in Figure 11(d1,d2), there were many irregular pits on the coating surface, indicating that the coating experienced fatigue wear. According to Figure 11(c1,c2,d1,d2), the addition of ceramic particles could change the wear mechanism of the coating, reducing the amount of wear and improving the wear resistance. At the same time, due to the role of fine grain strengthening and diffusion strengthening, the coating hardness value was improved, which was positively correlated with the wear resistance. Finally, the fine grain peening also increased the toughness of the coating, which in turn enhanced its wear resistance.

## 4. Conclusions

In this study, three groups of CoCrNi/TiC/TiB2 composite coatings were prepared by means of laser cladding technology, and their microstructures and properties were thoroughly investigated. Based on the findings, the main conclusions can be drawn as follows:
The coatings produced via laser cladding possessed good morphology characteristics without obvious pores and cracks, and the cladding layer showed strong metallurgical bonding with the matrix.The microstructure of the CoCrNi coating was primarily a combination of dendrites and equiaxed crystals, and the rapid solidification upon the laser cladding resulted in a uniform distribution of constituent elements. After the addition of ceramic particles, the grains were refined. The higher the amount of ceramic particles, the smaller the grains. The addition of ceramics transformed the CoCrNi-xTiC-xTiB2 composite coating from a single FCC phase to an FCC+TiC+TiB2 multiphase, and the diffraction peaks corresponding to TiC and TiB2 were subsequently enhanced when the ceramic content was increased from 10 wt.% to 30 wt.%.The average microhardness value of CoCrNi coatings increased by 16.37% compared with that of the substrate due to the fine grain strengthening. At 10 wt.% and 30 wt.% ceramic additions, the average microhardness values of the coatings were 328.02 HV and 494.77 HV respectively, an increase of approximately 82.02% and 174.55% compared to the substrate. This was due to the joint effect of fine grain strengthening and diffusion strengthening.The wear resistance of the coatings was positively correlated with the microhardness. Once the microhardness value increased, the wear amount decreased continuously. The substrate and the three sets of coatings experienced weight losses of 0.0088 g, 0.0082 g, 0.0038 g, and 0.0012 g, respectively. The addition of coating caused the change in wear mechanism of the material: while the substrate underwent adhesive, abrasive and fatigue wear, the pure CoCrNi coating suffered from adhesive and abrasive wear. Finally, the coatings with 10 wt.% and 30 wt.% ceramic contents experienced the adhesive and pitting fatigue wear mechanisms, respectively.

## Figures and Tables

**Figure 1 materials-16-07016-f001:**
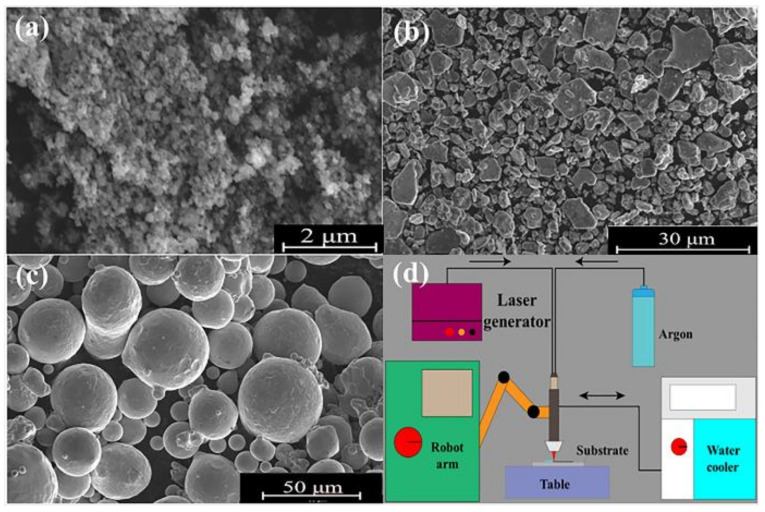
(**a**) TiC; (**b**) TiB2; (**c**) CoCrNi; (**d**) schematic diagram of experimental equipment.

**Figure 2 materials-16-07016-f002:**
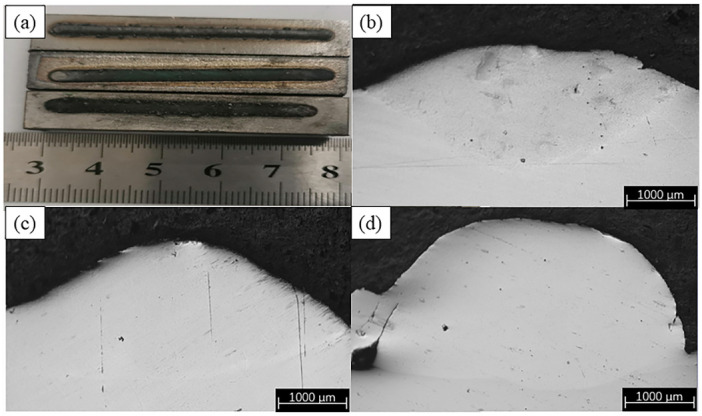
(**a**) Surface appearance of the coating. (**b**) Cross-sectional shape of TCB0. (**c**) Cross-sectional shape of TCB10. (**d**) Cross-sectional shape of TCB30.

**Figure 3 materials-16-07016-f003:**
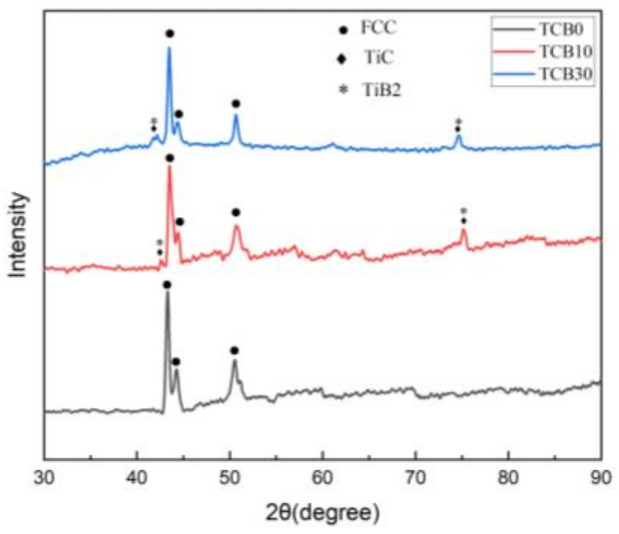
XRD patterns of coatings.

**Figure 4 materials-16-07016-f004:**
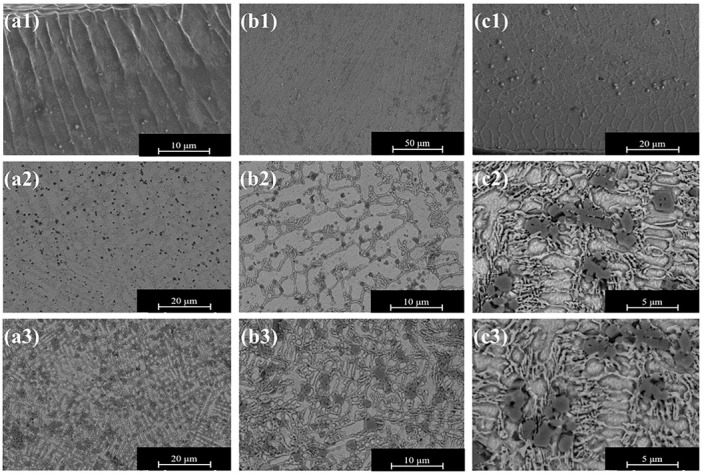
Coating microstructure: (**a1**)–(**c1**) TCB0; (**a2**)–(**c2**) TCB10; (**a3**)–(**c3**) TCB30.

**Figure 5 materials-16-07016-f005:**
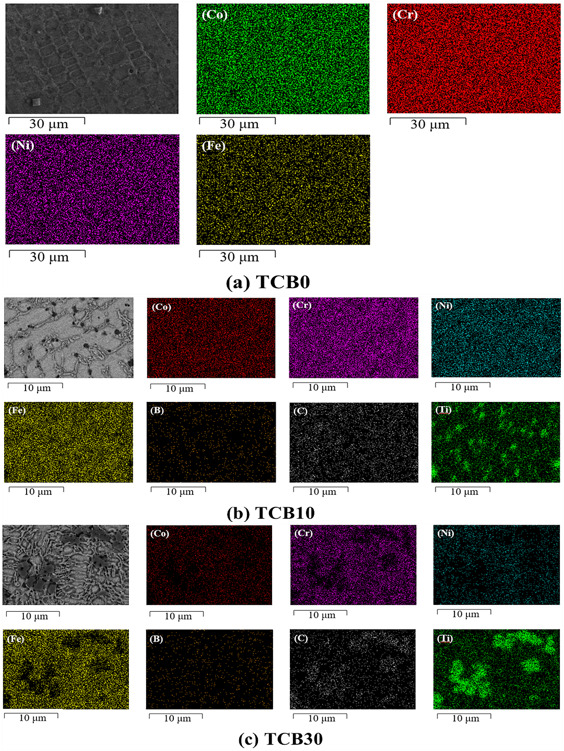
Elemental distribution diagrams. (**a**) TCB0; (**b**) TCB10; (**c**) TCB30.

**Figure 6 materials-16-07016-f006:**
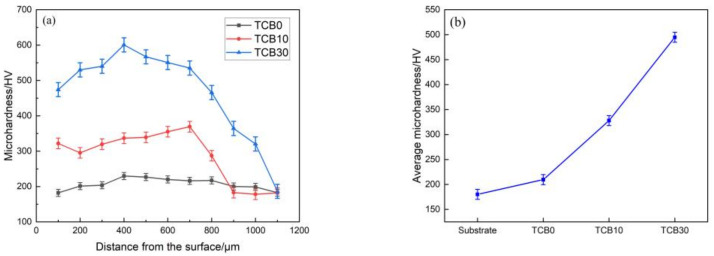
(**a**) Microhardness as a function of distance; (**b**) average microhardness.

**Figure 7 materials-16-07016-f007:**
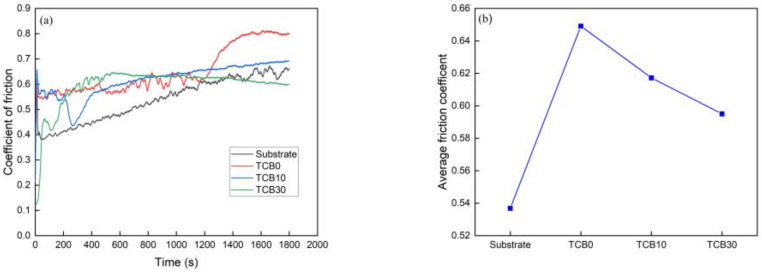
(**a**) Trend of friction coefficient; (**b**) average friction coefficient.

**Figure 8 materials-16-07016-f008:**
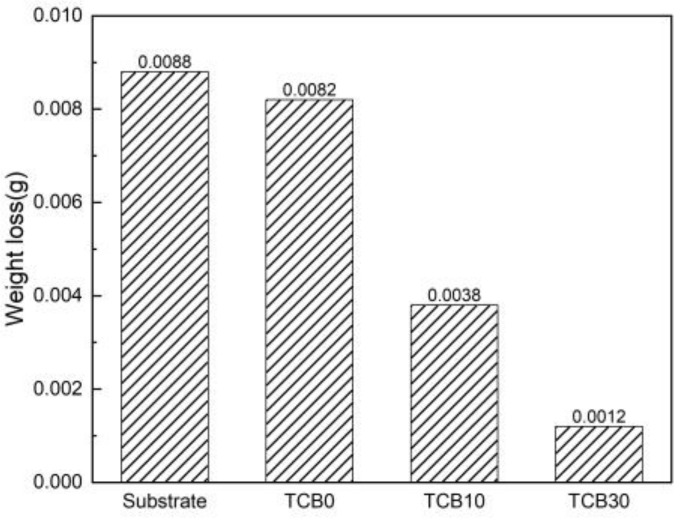
Weight loss of the substrate and coatings.

**Figure 9 materials-16-07016-f009:**
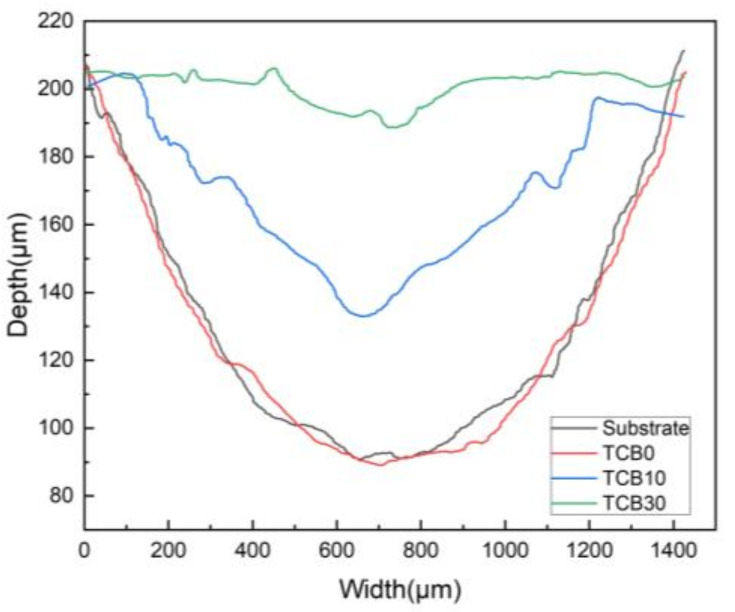
Wear cross-section morphology.

**Figure 10 materials-16-07016-f010:**
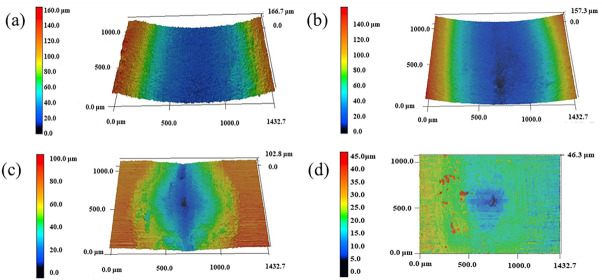
3D morphology plots of wear surface: (**a**) substrate; (**b**) TCB0; (**c**) TCB10; (**d**) TCB30.

**Figure 11 materials-16-07016-f011:**
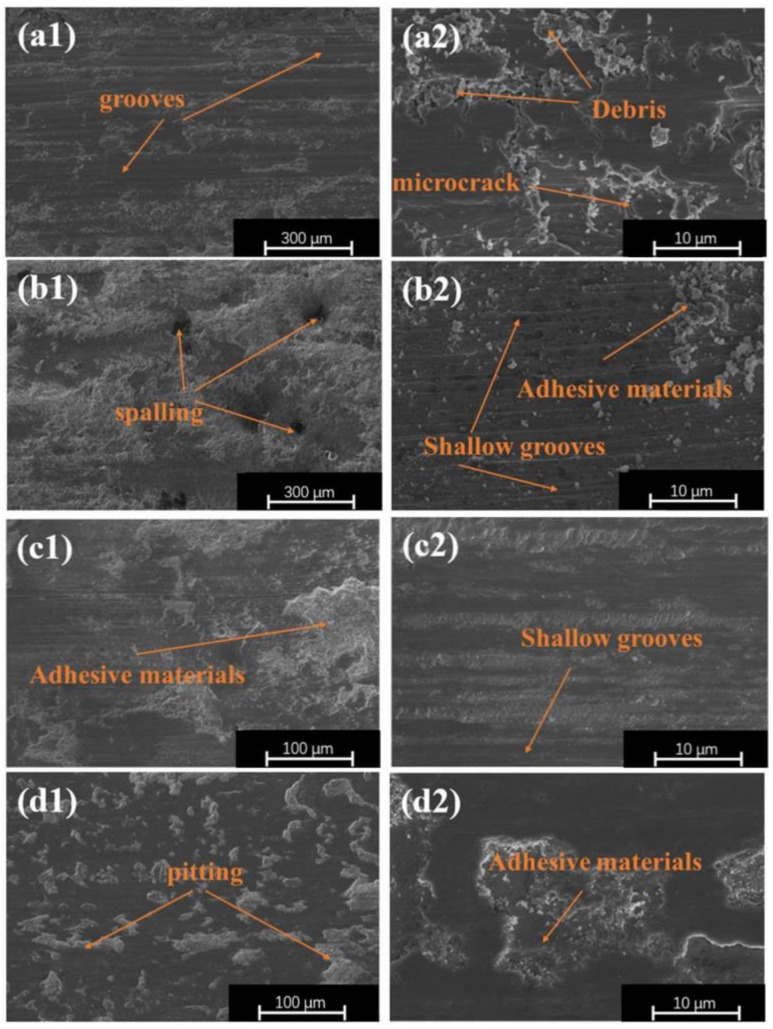
Wear patterns of the substrate and coatings: (**a1**,**a2**) substrate; (**b1**,**b2**) TCB0; (**c1**,**c2**) TCB10; (**d1**,**d2**) TCB30.

**Table 1 materials-16-07016-t001:** Laser cladding parameters.

Laser Power (W)	Scanning Speed (mm/s)	Spot Diameter (mm)	Protective Gas Flow (L/min)
1000	10	4	10

**Table 2 materials-16-07016-t002:** Coating number and ceramic content.

Coating Number	Powder Ratio
TCB0	100%CoCrNi
TCB10	90%CoCrNi + 5%TiC + 5%TiB2
TCB30	70%CoCrNi + 15%TiC + 15%TiB2

**Table 3 materials-16-07016-t003:** Coating dilution rate.

Coating Number	Depth of Cladding	Height of Cladding	Dilution Ratio
TCB0	250.86 μm	730.18 μm	25.57%
TCB10	255.70 μm	544.70 μm	31.95%
TCB30	499.55 μm	344.81 μm	59.16%

**Table 4 materials-16-07016-t004:** EDS results(wt.%).

	Co	Cr	Ni	Fe	B	C	Ti
TCB0	30.34	32.02	30.29	7.34			
TCB10	16.88	20.97	19.59	25.91	7.92	5.56	3.19
TCB30	5.47	17.93	8.13	27.84	16.08	13.53	11.01

**Table 5 materials-16-07016-t005:** Wear surface roughness parameters.

Samples	Ra	Rq	Rsk	Rku
Substrate	31.009	36.422	0.7345	2.2999
TCB0	29.878	35.295	0.7542	2.4443
TCB10	17.700	20.569	−0.2888	1.9766
TCB30	3.244	4.438	0.3996	5.2368

## Data Availability

The data that support the findings of this study are available from the corresponding author upon reasonable request.

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
