# Peer review of "Microstructure and Properties of CoCrNi/Nano-TiC/Micro-TiB2 Composite Coatings Prepared via Laser Cladding"

_materials, 2023, doi:10.3390/ma16217016_

Round 1
Reviewer 1 Report
Comments and Suggestions for Authors
The abstract is relatively weak for specific reasons;
(1) Are these "CoCrNi/TiC/TiB2 composite coating" all included simultaneously, or one by one?
(2) What is wt% you have used of the above contents, not mentioned in the abstract?
(3) Kindly check this statement "The main phases composing the coating were FCC, TiC and TiB2."
(4) "the addition of 30 wt.% ceramics'; what type of ceramic particles are you considering from the above-mentioned?
(5) fine grain peening, diffusion peening: are these really the keywords, which are even not appeared in the abstract?
(6) These types of clustering are not recommended [1][2][3][4], [7][8][9][10], [11][12][13][14] etc.
(7) Throughout the introduction I found only the addition of "CoCrNi", What about the other selected ceramic particles?
(8) I didn't find the exact objective and purpose of adding "CoCrNi/TiC/TiB2" in the introduction and for this study.
(9) How did you obtain the result of Table 3? Also, highlight the procedure in Section heading 2.
(10) The counter description of reading 3.2 XRD is totally insufficient. Kindly expand this.
(11) How do the authors perform the wear testing, on what kind of equipment, and against what parameters?
(12) Figure 9 is not understandable, what authors try to present here in terms of depth and width? also, write the units.
(13) The wear mechanism is not clear, relate the written understanding, description, and mechanism with Figure 11. Also highlight the regions of adhesive, abrasive wear, or any other type at Fig. 11.
(14) Rewrite the conclusion exactly based on the finding in terms of mass loss, hardness, XRD, and wear mechanism, and conclude based on the wt% of added particles.
Comments on the Quality of English Language
Minor changes can be enough
Reviewer 2 Report
Comments and Suggestions for Authors
The paper explores the performance of the cladding process on three different coatings, and overall, the research is well-structured, with clearly described methods and results. However, there are two key areas that the reviewer suggests addressing to better emphasize the importance of this research:
Major revision
Laser Process Parameters: It's not clear how the authors established the laser process parameters. Were these parameters derived from the existing literature or based on a preliminary campaign? Additionally, it would be beneficial to understand why the authors did not vary process parameters as a function of cladding. This information is crucial in contextualizing the research and its contribution.
Novelty and Contribution: Since the results of the study are expected to be influenced by the chemical composition of the coatings, it's essential to clarify the novelty of this paper. Furthermore, the paper should highlight whether the results confirm or refute findings from similar research in the field. A more explicit focus on these aspects would strengthen the significance of the research and its contribution to the existing body of knowledge.
Minor Revision:
Additionally, the reviewer notes that marker of Figure 4 is challenging to read. It would be advisable to improve the clarity of this figure to enhance the overall readability of the paper.
Addressing these major and minor revisions will help strengthen the paper's content and presentation, ensuring that the significance of the research is effectively communicated to the readers.
Round 2
Reviewer 1 Report
Comments and Suggestions for Authors
1-Authors still need to adjust the abstract, what they are writing must be concise, precise, and clear to the readers in terms of technical terms, methodology, and results trending.
2-For point 9 related to Table 3, include the explanation in the subsequent section for obtaining the dilution ratio.
3-Section 3.2 is not sufficient, as already pointed out in the first round; consider this critically.
4-For point 11, What type of equipment do you use, i.e. pin of disc, ring on disc?
5- Authors still need to take up the manuscript revision critically and seriously, still at many places critical consideration is required in terms of writing, and explanation of results like sections 3.1, 3.2, and 3.4.
Comments on the Quality of English Languageminor and moderate tuning is required.
Reviewer 2 Report
Comments and Suggestions for Authors
All comments have been discussed and paper can be accepted as is
Author Response
Thanks again for your comments on this manuscript